# Test and Analysis of Timekeeping Performance of Atomic Clock

**DOI:** 10.3390/s22249886

**Published:** 2022-12-15

**Authors:** Shuaichen Li, Chong Li, Jianfeng Wu, Haibo Cui

**Affiliations:** 1National Time Service Center, Chinese Academy of Sciences, College of Integrated Circuits, University of Chinese Academy of Sciences, Xi’an 710699, China; 2Beijing Satellite Navigation Center, Beijing 100085, China; 3National Time Service Center, Chinese Academy of Sciences, College of Electrical and Electronic Engineering, University of Chinese Academy of Sciences, Beijing 100049, China

**Keywords:** hydrogen atomic clock, cesium atomic clock, frequency stability, mobile timekeeping performance, atomic clock noise type

## Abstract

At present, there are few articles about the timekeeping performance of domestic atomic clocks in their moving state. In this paper, the frequency stability changes of hydrogen atomic and cesium atomic clocks in stationary and moving states are compared and analyzed; the frequency stability of the atomic clock at the beginning of its transition from moving state to stationary state is tested and analyzed; the influence of three main noises of atomic clocks on frequency stability is analyzed; and finally, the difference in the predictability of atomic clocks in moving and stationary states is analyzed. The results show that: (1) in the moving state, the frequency stability of a hydrogen clock decreases by 1–2 orders of magnitude, and the frequency stability of a cesium clock decreases by 0.5 orders of magnitude; (2) in the recovery stage, the frequency stability of hydrogen and cesium clocks is between that in static and moving stages, but the frequency stability fluctuates greatly in this stage; (3) in the moving state, the three main noises of the atomic clock all increase, of which the increase in the white noise of phase modulation is the largest, indicating that it is the most sensitive to vibration and has the greatest impact on the frequency stability of the atomic clock during the moving period; (4) in the mobile state, the RMS of the prediction data of the hydrogen clock and cesium clock greatly increases compared with that in the static state.

## 1. Introduction

In today’s highly information-based era, time and frequency are increasingly important in the military and civilian fields and have become very important resources [1]. The atomic clock is the core part of the timekeeping system. Its performance directly affects the stability of the timekeeping system. With recent developments and changes in the international situation, people’s demand for high performance and wide application environments in Chinese-made atomic clocks is increasing. In some specific use environments, such as military operations, there is a high demand for mobile punctuality. Therefore, the timekeeping performance of the atomic clock in its moving state is worth research and analysis.

At present, atomic clocks that are relatively mature in research and use mainly include hydrogen atomic clocks, cesium atomic clocks, and rubidium atomic clocks. At present, most timekeeping systems are made up of hydrogen atomic clocks and cesium atomic clocks. Hydrogen atomic clocks have excellent short-term stability and generally serve as the main clocks in the timekeeping system, while cesium atomic clocks have better medium- and long-term stability than hydrogen atomic clocks. They are generally used to remove the long-term drift of hydrogen atomic clocks [2]. They are used together in timekeeping according to their particular advantages. As the frequency source of precise time-frequency signal generation, the short-term frequency stability of the generated signal is determined by the performance of the frequency source itself and cannot be improved by compensation, while the long-term frequency stability can be compensated and improved by different control methods. Therefore, one of the bases for selecting the master clock is excellent short-term frequency stability. The hydrogen atomic clock has excellent short-term frequency stability and low noise characteristics. Therefore, the hydrogen clock is often used as the main clock [3].

At present, China’s hydrogen atomic clocks have been commercialized, with short-term frequency stability of up to $2\times 10^{-13}$ orders of magnitude. They are widely used in domestic timekeeping institutions and play an increasingly important role in the Beidou satellite navigation system [4]. At present, small, commonly used commercial cesium atomic clocks are divided into a magnetic separation state and optical pumping state. Early small cesium clocks were mainly in the magnetic separation state, with representative models including OSA3230B of the Osciquartz Company in Switzerland and 5071A of the Microsemi Company in the United States. The optically pumped small cesium clock does not have the beam optical system and multiplier of the magnetically separated small cesium clock. Instead, this is replaced by a laser, which reduces the difficulty of mechanical manufacturing, but requires a high level of electronic technology [5]. Many domestic institutions have made phased achievements in the research, development, and manufacture of magnetic-separation state and optically pumped small cesium clocks [6,7].

In terms of performance testing and analyzing China’s hydrogen atomic clock, Liu Fengyu [8] and others analyzed the long-term performance of China’s SOHM-4 hydrogen clock. The results show that the frequency accuracy is basically better than $5\times 10^{-13}$, and the linear relationship is not obvious. The frequency stability is between $1\times 10^{-14}$ and $2\times 10^{-14}$ based on the natural stability. Zhang Weiqun [9] and others analyzed the performance of the active hydrogen atomic clock at Shanghai Observatory. The performance test showed that the natural stability of the active hydrogen atomic clock was better than $2\times 10^{-15}$, and the temperature coefficient was better than $5\times 10^{-15}$/°C, which places it among the advanced ranks internationally. At present, a series of performance analyses has also been carried out for small cesium atomic clocks made in China, including frequency stability, frequency accuracy, and frequency drift under the static state of constant temperature and humidity. Compared with the 5071A cesium atomic clock made in the United States, the small cesium atomic clock made in China shows little difference in medium- and short-term frequency stability, but there is still a certain gap in accuracy and long-term frequency stability.

At present, research articles on the mobility of China’s hydrogen and cesium atomic clocks have not been reported. Therefore, this paper mainly analyzes the frequency stability of China’s hydrogen atomic clocks and China’s small cesium atomic clocks in the vehicle mobile state, including the stationary state, mobile state, and the stage of returning to a stationary state after moving (hereinafter referred to as the recovery stage), aiming at the research of hydrogen atomic clocks and cesium atomic clocks. The development and application provide a reference for performance indicators in the mobile state and provide a basis for further expanding the range of use of Chinese atomic clocks.

Whether the atomic clock can provide a long-term continuous and stable time-frequency signal is the most important condition for judging whether it can be used in the timekeeping system. Therefore, this paper will focus on the frequency stability index to evaluate the performance of the atomic clock. In many cases, due to the impact of the surrounding environment, it may not be possible to successfully collect data. At this time, the ability of the hydrogen atomic clock to predict future data based on past data is also required to be predictable. Therefore, predictability is also one of the indicators this paper is concerned about.

## 2. Test System and Test Method

The test system and method of atomic clock are described as follows.

### 2.1. Test System

This test was carried out on two mobile timekeeping cars. Some equipment of the timekeeping cars is shown in Figure 1. Each car was loaded with two hydrogen clocks and three cesium clocks. The temperature in the vehicle was 24 ± 4 °C; the hydrogen bell was equipped with a thermostat, which showed a temperature of 22 ± 0.1 °C; and the relative humidity was 40–60%. In the moving test, the moving path was the urban road. The speed when driving at a constant speed was about 50 km/h, and the driving time was about 8 h. The connections of some equipment of the timekeeping car are shown in Figure 1.

Since there was no traceability condition in the moving state of this test, and in order to improve the reliability of generating time-frequency signals, two hydrogen clocks and three cesium clocks were used as the main frequency sources to jointly generate precise time-frequency signals. The hydrogen atomic clock with 10,000 s frequency stability better than $5\times 10^{-15}$ was selected as the main clock in the static state, and the cesium clock with 10,000 s frequency stability better than $8.5\times 10^{-14}$ was selected as the main clock in the moving state. The switching of the main clock was completed using a lossless switch. First, the 10MHz clock signals of two hydrogen clocks and three cesium clocks of two vehicles were connected to the main and standby clock switching device; the main clock was selected and switched using software, and then the signals entered the phase micro-jump meter to generate 1PPS signals and 10MHz signals. Then, the 10MHz signals were sent to the phase comparator as reference signals, and the 1PPS signals were sent to the counter as reference signals. In order to increase the reliability and stability of the system, the system used a phase comparator and a counter to collect data at the same time, and the data collected by the two were mutually backed up.

### 2.2. Test Method

The most commonly used indicators to characterize the performance of atomic clocks are frequency accuracy, frequency drift rate and frequency stability. Whether the atomic clock can provide long-term continuous and stable time-frequency signal is the most important judgment condition for whether it can be used in a timekeeping system. Therefore, in many related studies, the performance evaluation of the atomic clock mainly focuses on the frequency stability index. Therefore, this paper mainly tests and analyzes the performance of atomic clocks in static state, moving state and recovery phase with frequency stability as an indicator.

The hydrogen clock stability test method used in this paper is shown in Figure 2. In order to distinguish each atomic hydrogen clock, they are named hydrogen clock A and hydrogen clock B.

In the moving state, it is impossible to find a time signal with a higher frequency stability than the hydrogen clock as a reference. Therefore, the two hydrogen clocks are self-evaluated; that is, we subtracted the clock difference data of the two hydrogen clocks, calculated the Allen variance of the new clock difference data and divided it by $\sqrt{2}$.

The stability test method of the cesium clock used in this paper is shown in Figure 3. In order to distinguish cesium atomic clocks, they are named cesium clock A, cesium clock B and cesium clock C.

We connected the 10MHz clock signal of the atomic clock to the phase comparator and counter, respectively, and completed the time difference and phase difference measurement with the main clock signal as the reference. The Allan variance was analyzed by time difference and phase difference data to analyze the frequency stability of the atomic clock.

### 2.3. Frequency Stability Analysis Method

The frequency stability of an atomic clock is the standard in measuring the stability of its signal frequency and is an important indicator in measuring the performance of the atomic clock [10]. One of the most common methods to analyze frequency stability is to calculate the Allen variance. The Allan variance method is a time-domain analysis method [11], and its standard calculation formula is:(1)$$\sigma_{y}\left(t\right)=\sqrt{\frac{1}{2(m-1)}\times \sum_{i=1}^{m-1}{\left(y_{i+1}-y_{i}\right)}^{2}}$$
where $y_{i}(i=1,2,3\cdots ,m)$ is the relative frequency measurement value at time i; $x_{i+1}$ is the relative frequency measurement value at time i+1; and m is the number of measurement samples, which are equally spaced.

At the same time, the Allan variance formula can also be calculated using the phase measurement value. The calculation formula is:(2)$$\sigma_{y}\left(\tau \right)=\sqrt{\frac{1}{2(N-2)\tau^{2}}\sum_{i=1}^{N-1}{\left(x_{i+2}-2x_{i+1}+x_{i}\right)}^{2}}$$
where $x_{i}(i=1,2,3\cdots ,m)$ is the phase measurement value at time i; $x_{i+1}$ is the phase measurement value at time i+1; $x_{i+2}$ is the phase measurement value at time i+2; and N is the number of measurement samples, which are equally spaced.

### 2.4. Noise Type of Atomic Clock

The noise of atomic clock can be seen as the superposition of five kinds of noise, namely: white phase modulation (WPM), flicker phase modulation (FPM), white frequency modulation (WFM), flicker frequency modulation (FFM) and random walk frequency modulation (RWFM) [12]. Among them, white phase modulation noise, white FM noise and random walk FM noise have a large impact on the frequency stability of atomic clock [13]. The error of an atomic clock is divided into two parts: deterministic error and random noise. Deterministic deviation can be compensated by model prediction, and the random noise part usually uses Allen variance or the power law spectrum model of noise to estimate the noise strength [14]. The main feature of the Allan variance method is that it is very easy to characterize and identify various error sources and their contributions to the overall noise statistical characteristics in detail, and it is also easy to calculate, separate, and identify and quantify different noise items in the data. The power law spectrum curve of various noises of cesium clock characterized by the square root value of Allan variance is shown in Figure 4.

The relationship between Allen variance and power spectral density is:(3)$$\sigma_{y}^{2}\left(\tau \right)=2\int_{-\infty }^{\infty }S_{y}(f)\frac{\sin^{4}\left(\pi f\tau \right)}{{\left(\pi f\tau \right)}^{2}}df=4\int_{0}^{\infty }S_{y}(f)\frac{\sin^{4}\left(\pi f\tau \right)}{{\left(\pi f\tau \right)}^{2}}df$$
where $S_{y}(f)$ is the power spectral density.

The relationship between white phase modulation noise, white FM noise, random walk FM noise and Allen variance is introduced in detail in the literature [15]:(4)$$\sigma_{WPM}^{2}=3\frac{\sigma_{1}^{2}}{\tau }$$
(5)$$\sigma_{WFM}^{2}=\frac{\sigma_{2}^{2}}{\tau }$$
(6)$$\sigma_{RWFM}^{2}=\frac{1}{3}\sigma_{3}^{2}\tau$$

Different types of noise of the atomic clock can be identified from the Allen variance curve. The slope of white phase modulation noise (WPM) in logarithmic Allan chart is—2, the slope of white frequency modulation noise (WFM) in logarithmic Allan chart is—1, and the slope of random walk frequency modulation noise (RWFM) in logarithmic Allan chart is 1. The value of WPM can be read at $\tau =\sqrt{3}$ in the logarithmic Allan graph, the value of WFM can be read at τ=1 in the logarithmic Allan graph, and the value of RWFM can be read at τ=3 in the logarithmic Allan graph.

### 2.5. Predictability of Atomic Clock

In many cases, due to the impact of time and environment, it is impossible to collect data smoothly. At this time, it is necessary to predict the missing data based on the previously collected data, which also requires the predictability of Chinese atomic clocks in the mobile state. The purpose of this section is to compare and analyze the difference between the predictability of Chinese atomic clocks in static and mobile states.

This section builds a mathematical model based on the measurement data collected by the phase comparator in the past and quantifies the predictability of the atomic clock by calculating the difference between the fitting value and the real value, that is, residual error and RMS. The prediction of atomic clock data can be roughly divided into clock error prediction and clock speed prediction. This section uses a clock error prediction method to analyze the predictability of atomic clocks. Common clock error prediction methods include least square method, FIR filter and Kalman filter. Kalman filtering is suitable for application when the clock difference state model is linear and the statistical characteristics of state noise and measurement noise are known, and the least squares method requires less data. Therefore, this section uses linear fitting and quadratic fitting methods to predict the clock error data at the next time based on the previous measurement data and calculates the difference between the fitting value and the real value, i.e., residual, to evaluate the predictability of the atomic clock.

Linear fitting means that given a series of $x_{i}$, $y_{i}(i=1,2,\cdots N)$, you can use y=kx+b to fit the data linearly, so that $x_{i}$, $y_{i}$ and the straight line are as close as possible. The residual between the actual measured value and the fitted value can be expressed as:(7)$$f=\sum {\left(y_{i}-kx_{i}-b\right)}^{2}$$

The greater the residual error, the better the fitting effect. Therefore, the problem is transformed into solving k and b to minimize f, where a is a function of two variables, and in order to obtain the minimum value of f, the first derivative of f with respect to k and b is 0, so:(8)$$\frac{\partial f}{\partial b}=2\sum_{i=1}^{N}(kx+b-y_{i})=0$$
(9)$$\frac{\partial f}{\partial k}=2\sum_{i=1}^{N}(kx+b-y_{i})x_{i}=0$$

By sorting out Equation (9), it can be concluded that the equation to be satisfied by the fitting curve is:(10)$$\left\{ \begin{array}{l} (\sum_{i=1}^{N}x_{i})b+(\sum_{i=1}^{N}x_{i}^{2})k=\sum_{i=1}^{N}x_{i}y_{i}\\ Nb+(\sum_{i=1}^{N}x_{i})k=\sum_{i=1}^{N}y_{i} \end{array}\right.$$

Quadratic fitting means that given a series of $x_{i}$, $y_{i}(i=1,2,\cdots N)$, you can use $y=ax^{2}+bx+c$ to perform quadratic fitting on the data, so that $x_{i}$, $y_{i}$ and the conic curve are as close as possible. The residual between the actual measured value and the fitted value can be expressed as:(11)$$f=\sum (y_{i}-ax_{i}^{2}-bx_{i}-c)$$

The greater the residual error, the better the fitting effect. Therefore, the problem is transformed into solving a, b, and c to minimize f, where f is a function of three variables, and the first derivative of f with respect to a, b and c is 0 to obtain the minimum value of f, so:(12)$$\left\{ \begin{array}{l} \frac{\partial f}{\partial a}=2\sum_{i=1}^{N}(y_{i}-ax^{2}-bx-c)x_{i}^{2}=0\\ \frac{\partial f}{\partial b}=2\sum_{i=1}^{N}(y_{i}-ax^{2}-bx-c)x_{i}=0\\ \frac{\partial f}{\partial c}=2\sum_{i=1}^{N}\left(y_{i}-ax^{2}-bx-c\right)=0 \end{array}\right.$$

## 3. Test Results and Analysis

The main work of this paper is as follows: to use the phase comparator to collect the clock error data of the atomic clock for three consecutive days to compare and analyze the difference between the frequency stability of the atomic clock in the stationary state and the mobile state; to test and analyze the frequency stability of the atomic clock in the recovery phase; to test and analyze the three main noises of the atomic clock in static and moving states; and to compare any analyze the predictability of atomic clocks in stationary and moving states. The idea of this paper is presented in the form of a figure, as shown in Figure 5.

### 3.1. Frequency Stability in Static State

The nominal indexes of hydrogen atomic clock under static state are shown in Table 1.

First, the frequency stability of the hydrogen clocks on the two vehicles under static state is tested. The Allen variance of the clock difference data collected by the phase comparator for three consecutive days is calculated, as shown in Table 2 and Figure 6.

From the data in Table 2, we can see that the frequency stability of the hydrogen clocks of the two cars is similar, and car 1 is slightly better than car 2. In the 1 s–100 s period, the frequency stability of the hydrogen clock can reach $10^{-14}$. In the 100 s–10,000 s period, the frequency stability of the hydrogen clock can reach $10^{-15}$, which basically meets the performance requirements of the frequency stability of the hydrogen clock.

Next, the frequency stability of the six cesium clocks on the two cars at stationary state were tested. The Allan variance of the clock difference data from the atomic clocks collected by the phase comparator for 3 consecutive days was calculated, and the results are shown in Table 3 and Figure 7 and Figure 8.

From the data in Table 3, it can be seen that the frequency stability of the six cesium clocks is similar. The frequency stability of the cesium clocks can reach $10^{-12}$ magnitude in the 1 s–10 s time period, and the frequency stability of the hydrogen clocks can reach $10^{-13}$ magnitude in the 100 s–10,000 s time period, which basically meets the performance requirements of the frequency stability of the cesium clocks.

As the frequency stability of the hydrogen and cesium atomic clocks on board car 1 is similar to the performance of the atomic clocks on board car 2, the subsequent aim of this paper is to only process and analyze the data from car 1.

### 3.2. Frequency Stability of Atomic Clocks in the Moving State

This section tests and analyzes the frequency stability of hydrogen and cesium clocks in the moving state. The Allan variance is calculated for approximately 8 h of continuous atomic clock differential data collected by the phase meter. The calculated frequency stability of the two hydrogen atomic clocks in the moving state for mutual evaluation is shown in Table 4 and Figure 9.

Figure 10 shows a comparison of the frequency stability of the hydrogen clock in the stationary and moving states.

As can be seen from the data in Table 4, the frequency stability of the two hydrogen clocks is similar in the moving state. In the 10 s–100 s period, the frequency stability of the hydrogen clocks in the moving state can reach an order of magnitude of $10^{-12}$, and the corresponding frequency stability in the stationary state can reach an order of magnitude of $10^{-14}$; in the 1000 s–4000 s period, the frequency stability of the hydrogen clocks in the moving state can reach an order of magnitude of $10^{-13}$, and the corresponding frequency stability in the stationary state can reach an order of magnitude of $10^{-14}∼10^{-15}$.

Comparing the Allan’s variance of the two hydrogen clocks at rest and in motion shows that movement decreases the frequency stability of the hydrogen clock by an order of magnitude of 1∼2 from 1–4000 s, so the effect of movement on the frequency stability of the hydrogen clock is approximately an order of magnitude of 1∼2.

The results of the frequency stability calculations for the cesium clock are shown in Table 5 and Figure 11.

Figure 12 shows a comparison of the frequency stability of the cesium clock in the stationary and moving states.

From the data in Table 5, it can be seen that the frequency stability of the three cesium clocks is similar in the moving state. In the period of 1 s–10 s, the frequency stability of the cesium clock in the moving state can reach an order of magnitude of $10^{-11}∼10^{-12}$, and the corresponding frequency stability in the stationary state can reach an order of magnitude of $10^{-12}$; in the period of 1000 s–4000 s, the frequency stability of the hydrogen clock in the moving state can reach an order of magnitude of $10^{-12}∼10^{-13}$, and the corresponding frequency stability in the stationary state can reach an order of magnitude of $10^{-13}$.

Comparing the Allan variance of the three cesium clocks at rest and in motion shows that movement deteriorates the frequency stability of the cesium clocks by about one order of magnitude from 1–10 s and by about 0.5 orders of magnitude from 10–4000 s. That is, the effect of movement on the frequency stability of the cesium clock is approximately an order of magnitude of 0.5∼1.

The short-term stability of the phase micro-jump meter used in this test and the short-term stability of the lossless switch are both far less than the short-term stability of the cesium clock in the moving state, so the impact of the phase micro-jump meter and the lossless switch is ignored.

### 3.3. Frequency Stability in the Recovery Phase

Approximately 16 h of atomic clock differential data were collected after the timekeeping vehicle had switched from the moving state to the stationary state. This part of the work was aimed at testing and analyzing the frequency stability of the atomic clock during the recovery phase.

The results of the calculation of the frequency stability of the hydrogen clock are shown in Table 6 and Figure 13.

Figure 14 shows a comparison of the frequency stability in the three states of hydrogen clock A and hydrogen clock B.

As can be seen in Figure 14, the Arrhenius variance of the hydrogen clock differential data during the recovery from the moving process to the stationary process lies between the Arrhenius variance of the stationary and moving states, and the short-term stability is of the same order of magnitude as the Arrhenius variance in the stationary state. The frequency stability of the hydrogen atomic clock can reach $2\times 10^{-13}$ for a period of time after the end of the movement and the return to rest, but the fluctuations in frequency stability during this phase are large. The characteristics of the recovery phase are that the short-term stability recovers quickly, but the long-term stability fluctuates greatly.

The results of the cesium clock are shown in Table 7 and Figure 15.

Figure 16 shows a comparison of the frequency stability of the cesium clock in three states: stationary, moving and recovery phases.

As can be seen in Figure 16, during the recovery from the moving process to the stationary process, the Allan variance of the cesium clock differential data is between the Allan variance of the stationary and moving states, and the frequency stability is of the same order of magnitude as the Allan variance of the stationary state. Cesium atomic clocks can achieve a short-term frequency stability of $3\times 10^{-12}$ and a long-term frequency stability of $1\times 10^{-12}$ during the recovery phase.

## 4. Analysis of the Effects of Different Types of Noise in Atomic Clocks

This section analyses the effect of the three main types of noise on the frequency stability of an atomic clock. The aim of this part of the work is to test and analyze the three main types of noise in the atomic clock at rest and in motion, to observe which noise increases the most during motion, and thus to speculate which noise has the greatest effect on the frequency stability of the atomic clock during motion.

The results of the comparison of the three main noises of the hydrogen atomic clock at rest and in motion are shown in Table 8.

Table 8 shows that the three main types of noise in the moving state of the hydrogen atomic clock increase compared to the stationary state. The white noise frequency increases by about 2 orders of magnitude, the random wander noise frequency increases by 1 order of magnitude, and the phase-modulated white noise increases by about 3–4 orders of magnitude. During the moving process, the increase in phase-modulated white noise is the largest, and it is assumed that the noise that has the greatest effect on the frequency stability of the hydrogen clock during the moving process is the phase-modulated white noise.

The results of the comparison of the three main types of noise in the stationary and moving states of the cesium atomic clock are shown in Table 9.

In Table 9, it can be seen that the three main types of noise of the cesium atomic clock increase in the moving state compared to the stationary state. The increases in white noise frequency and random wander noise frequency are very small, increasing by 0.5 orders of magnitude, and the increase in phase-modulated white noise is about 3–4 orders of magnitude. During the moving process, the increase in phase-modulated white noise is the largest, and it is assumed that the noise that has the greatest effect on the frequency stability of the cesium clock during its movement is the phase-modulated white noise.

## 5. Predictability Results and Analysis

This section quantitatively assesses the predictability of atomic clocks using linear and quadratic fitting methods. The differential seconds of clock data for atomic clocks at rest and in motion, collected using a phase ratio meter, were used.

A total of 35,000 sets of hydrogen clock data were collected. The first 30,000 sets of data were used as a training set, and the second 5000 sets were used as a test set for the prediction results. The residuals and RMS of the latter 5000 sets of predicted and real data were calculated. The results of the hydrogen clock implementation are shown in Figure 17, Figure 18, Figure 19, Figure 20, Figure 21 and Figure 22.

Figure 17 shows the results of the linear fit of the clock difference data for the stationary state of hydrogen clock A. The black line represents the real data, and the red line represents the linear fit. From Figure 17, it can be seen that the linear model forecast curve is basically consistent with the trend of the real clock difference data curve, and the forecast data have a high overlap with the measured data.

Figure 18 shows the results of the linear fit of the clock difference data in the moving state of hydrogen clock A. The black line represents the real data, and the red line represents the linear fit results.

Figure 19 shows a comparison of the residuals of the linear fit for the stationary and moving states of hydrogen clock A. The black line represents the residuals for the stationary state, and the red line represents the residuals for the moving state. As can be seen in Figure 19, the residuals of the linear fit in the stationary state are much smaller than those of the linear fit in the moving state.

Figure 20 shows the results of the quadratic fit of the clock difference data for the stationary state of hydrogen clock A. The black line represents the real data, and the red line represents the linear fit. As can be seen in Figure 20, the linear model forecast curve is almost identical to the real clock difference data curve, and the overlap between the forecast data and the real data is very high.

Figure 21 shows the results of the quadratic fit of the clock difference data for the moving state of hydrogen clock A. The black line represents the real data, and the red line represents the linear fit. As can be seen in Figure 21, the linear model forecast curve differs significantly from the real clock difference data curve, and the overlap between the forecast data and the real data is low.

Figure 22 shows a comparison of the residuals of the quadratic fit for the stationary and moving states of hydrogen clock A. The black line represents the residuals in the stationary state, and the red line represents the residuals in the moving state. As can be seen in Figure 22, the residuals of the quadratic fit in the stationary state are smaller than the residuals of the linear fit in the moving state. The residuals of the quadratic fit for the stationary hydrogen atomic clock are very close to zero, indicating that the Chinese-made hydrogen atomic clock is highly predictable in the stationary state.

The residuals and RMS of the linear and quadratic fits to the hydrogen clock for the stationary and moving states are shown in Table 10.

As can be seen in Table 10, the quadratic fit is significantly better than the linear fit in both the stationary and moving states. The residuals and RMS of the linear fit increased from 0.28 to 5.33 in the moving state, while the residuals and RMS of the quadratic fit increased from 0.03 to 2. This shows that the predictability of the Chinese hydrogen atomic clocks used in this test deteriorated significantly in the moving state compared to the stationary state, with the Chinese hydrogen atomic clocks in the moving state having little predictability. The reasons for this decrease in predictability include the unpredictability of unexpected road conditions that may be encountered by the timekeeping car during movement and the large fluctuations in the differential clock data during movement.

There are 33,000 sets of cesium clock data. Of these, the first 30,000 sets of data were used as the data training set, and the second 3000 sets of data were used as the prediction result validation set. The residuals and RMS of the latter 3000 sets of predicted and real data were calculated. The results of the cesium clock implementation are shown in Figure 23, Figure 24, Figure 25, Figure 26 and Figure 27.

Figure 23 shows the results of the linear fit of the clock difference data for the stationary state of cesium clock A. The black line represents the real data, and the red line represents the linear fit. In Figure 23, it can be seen that the linear model forecast curve is less different from the real clock difference data curve, and the forecast data have a high overlap with the measured data.

Figure 24 shows the linear fit results of the clock difference data for the moving state of cesium clock A. The black line represents the real data, and the red line represents the linear fit results. As can be seen in Figure 24, the linear fit model forecast curve differs significantly from the real clock difference data curve, and the overlap between the forecast data and the real data is low.

Figure 25 shows the comparison of the residuals of the linear fit for the stationary and moving states of cesium clock A. The black line represents the residuals for the stationary state, and the red line represents the residuals for the moving state. In Figure 25, it can be seen that the residuals of the linear fit in the stationary state are smaller than those of the linear fit in the moving state.

Figure 26 shows the linear fit results of the clock difference data for the moving state of cesium clock A. The black line represents the real data, and the red line represents the linear fit results. As can be seen in Figure 26, the difference between the forecast curve of the quadratic fit model and the curve of the real clock difference data is small, and the forecast data have a high overlap with the measured data.

Figure 27 shows the results of the quadratic fit of the clock difference data for the moving state of cesium clock A. The black line represents the real data, and the red line represents the linear fit. From Figure 27, it can be seen that the quadratic fit model forecast curve differs greatly from the real clock difference data curve, and the overlap between the forecast data and the real data is low.

Figure 28 shows a comparison of the residuals of the quadratic fit for both the stationary and moving states of cesium clock A. The black line represents the residuals for the stationary state, and the red line represents the residuals for the moving state. As can be seen in Figure 28, the residuals of the linear fit in the stationary state are much smaller than the residuals of the linear fit in the moving state.

The residuals and RMS of the linear and quadratic fits of the cesium clock for the stationary and moving states are shown in Table 11.

As can be seen in Table 11, the quadratic fit was better than the linear fit in the stationary state, but the linear fit was significantly better than the quadratic fit in the moving state. The RMS of the linear fit in the moving state increased from 0.7938 to 4.06 in the stationary state, and the residuals and RMS of the quadratic fit increased from 0.1107 to 45 in the stationary state. The reasons for this decrease in predictability include the unpredictability of unexpected road conditions that may be encountered by the timekeeping vehicle during movement and the fluctuation of the differential data during the movement of the hydrogen atomic clock.

## 6. Summary of This Article

This paper compares and analyzes the difference in frequency stability between Chinese-made hydrogen and cesium atomic clocks in stationary and moving states. The data show that the effect of movement on the frequency stability of the hydrogen clock is of an order of magnitude of 1∼2, the short-term stability can reach an order of magnitude of $1\times 10^{-12}$, and the long-term stability can reach an order of magnitude of $10^{-13}$ in the moving state. The frequency stability of the cesium clock is reduced by 0.5 orders of magnitude, with short-term stability dropping to an order of magnitude of $10^{-11}∼10^{-12}$; the test analyzes the frequency stability of the atomic clock at the beginning of its transition from the moving state to the stationary state. The data show that the Arrhenius variance performance of the hydrogen and cesium clocks during the transition from the moving to the stationary process lies between the stationary and moving states, and the short-term stability is of the same order of magnitude as the Arrhenius variance in the stationary state, but the fluctuations in frequency stability are larger at this stage. The frequency stability of the hydrogen clock can reach an order of $2\times 10^{-13}$, and that of the cesium clock can reach $1\times 10^{-12}$. The analysis of the three main noises of the atomic clock shows that all three main noises increase in the moving state, among which, the increase in the phase-modulated white noise is the largest, indicating that the phase-modulated white noise is more sensitive to vibration and it has a greater impact on the frequency stability during the moving period. The RMS of the linear fit of the hydrogen clock increases to a large extent in the moving state compared to the stationary state. Compared with the performance of the laboratory, the static and dynamic indicators of the on-board hydrogen clock decreased to a certain extent, which is mainly affected by the vibration, temperature and humidity control, electromagnetic environment, power supply, etc. In this special application, the design and control measures in these aspects should be strengthened.

## Figures and Tables

**Figure 1 sensors-22-09886-f001:**
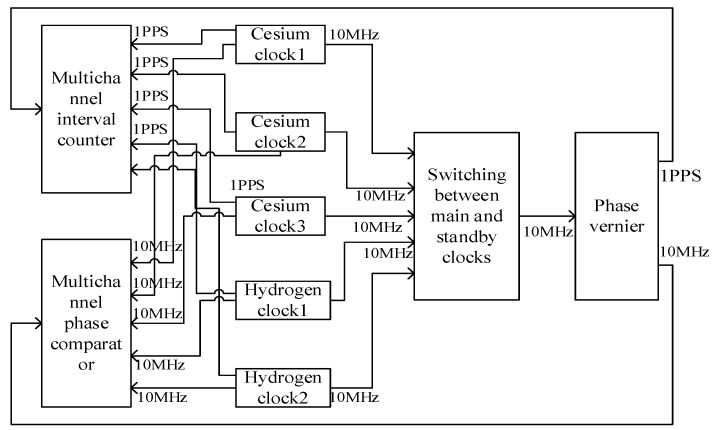
Connection Diagram of Some Equipment of Timekeeping System.

**Figure 2 sensors-22-09886-f002:**
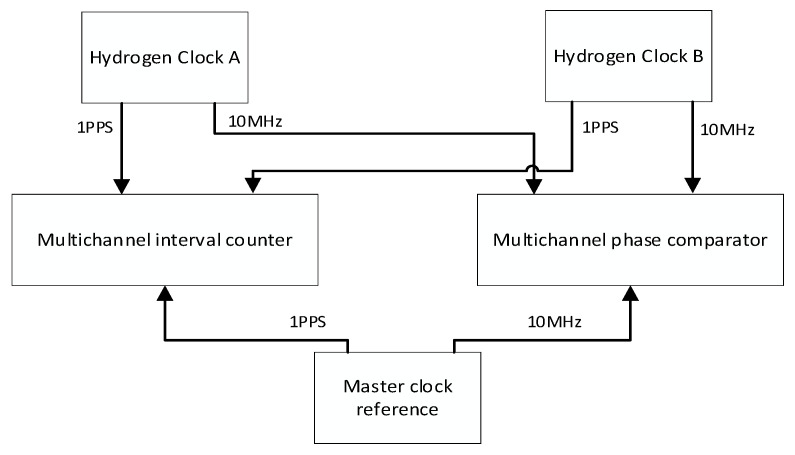
Frequency stability test method of hydrogen clock.

**Figure 3 sensors-22-09886-f003:**
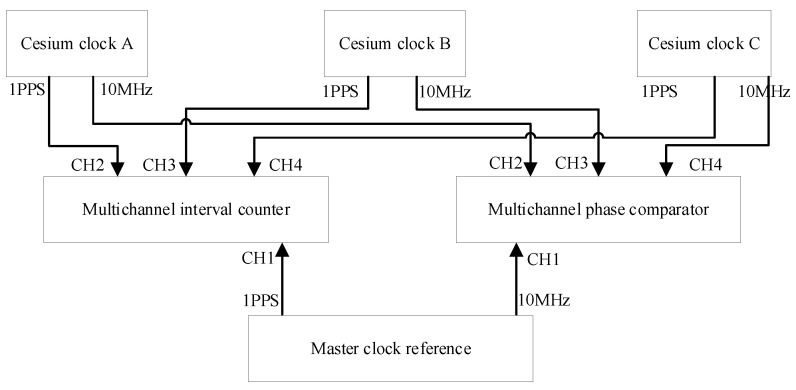
Frequency stability test method of cesium clock.

**Figure 4 sensors-22-09886-f004:**
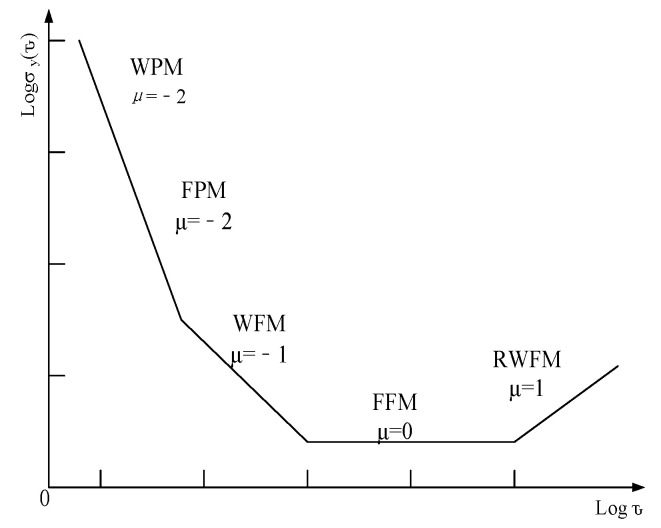
Noise Type of Atomic Clock.

**Figure 5 sensors-22-09886-f005:**
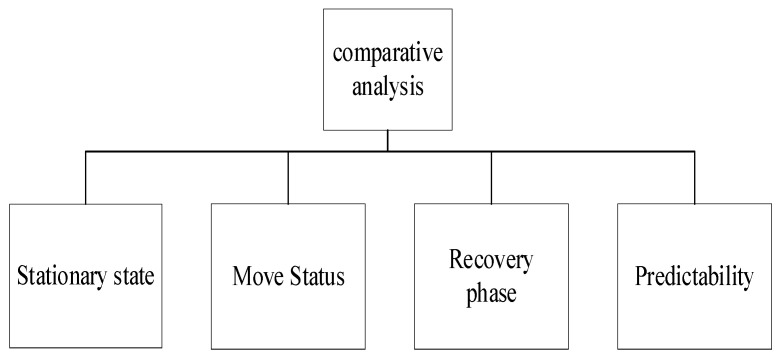
Comparative analysis of test data in this paper.

**Figure 6 sensors-22-09886-f006:**
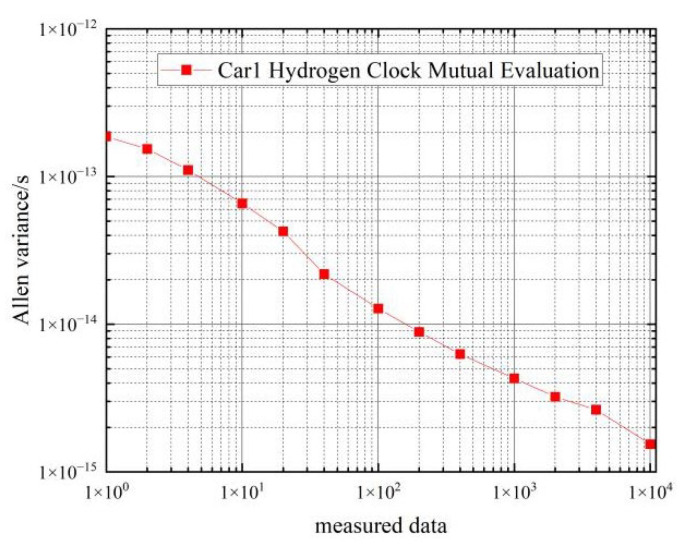
Frequency stability of car 1 hydrogen clock in a stationary state.

**Figure 7 sensors-22-09886-f007:**
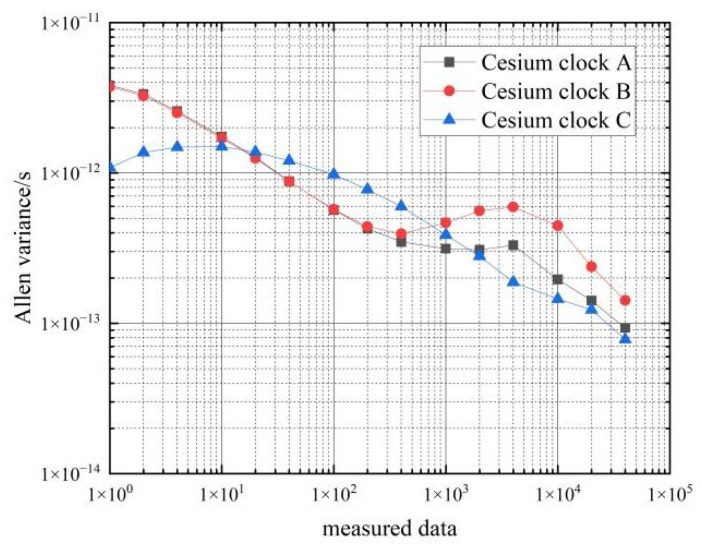
Frequency stability of the car 1 cesium clocks at stationary state.

**Figure 8 sensors-22-09886-f008:**
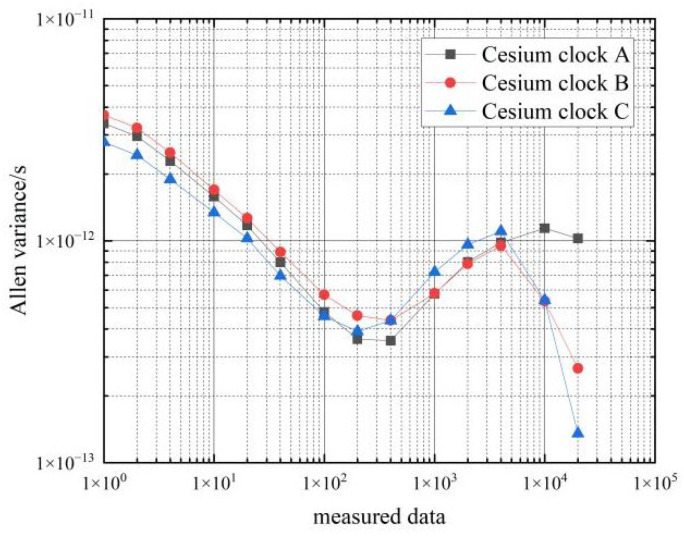
Frequency stability of the car 2 cesium clocks at stationary state.

**Figure 9 sensors-22-09886-f009:**
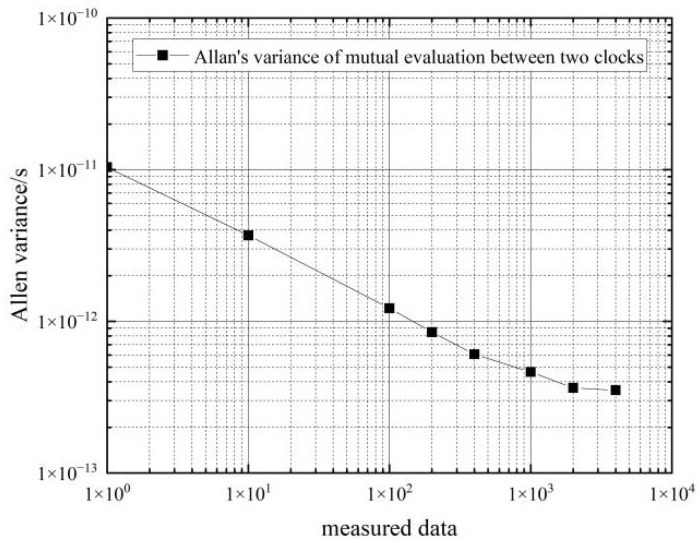
Frequency stability of two hydrogen clocks in the moving state for mutual evaluation.

**Figure 10 sensors-22-09886-f010:**
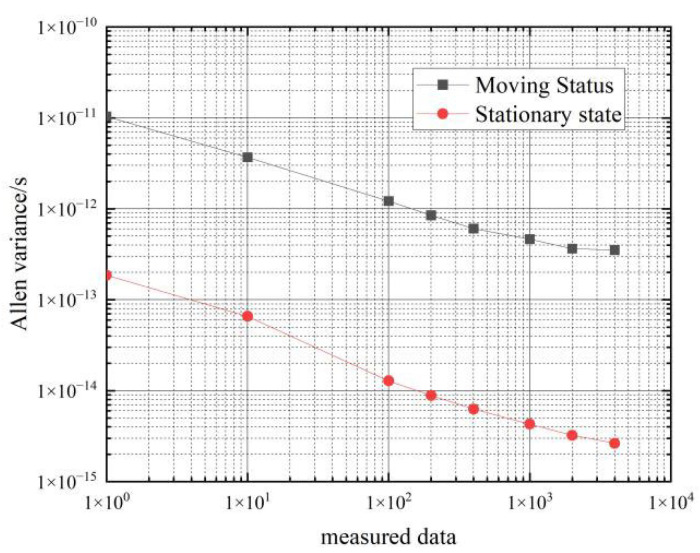
Comparison of the frequency stability of a cesium clock at stationary and moving states.

**Figure 11 sensors-22-09886-f011:**
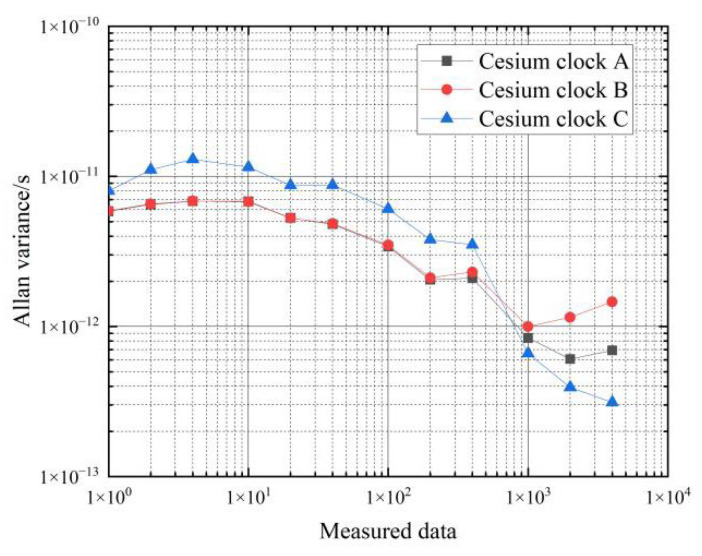
Frequency stability of cesium clock moving state.

**Figure 12 sensors-22-09886-f012:**
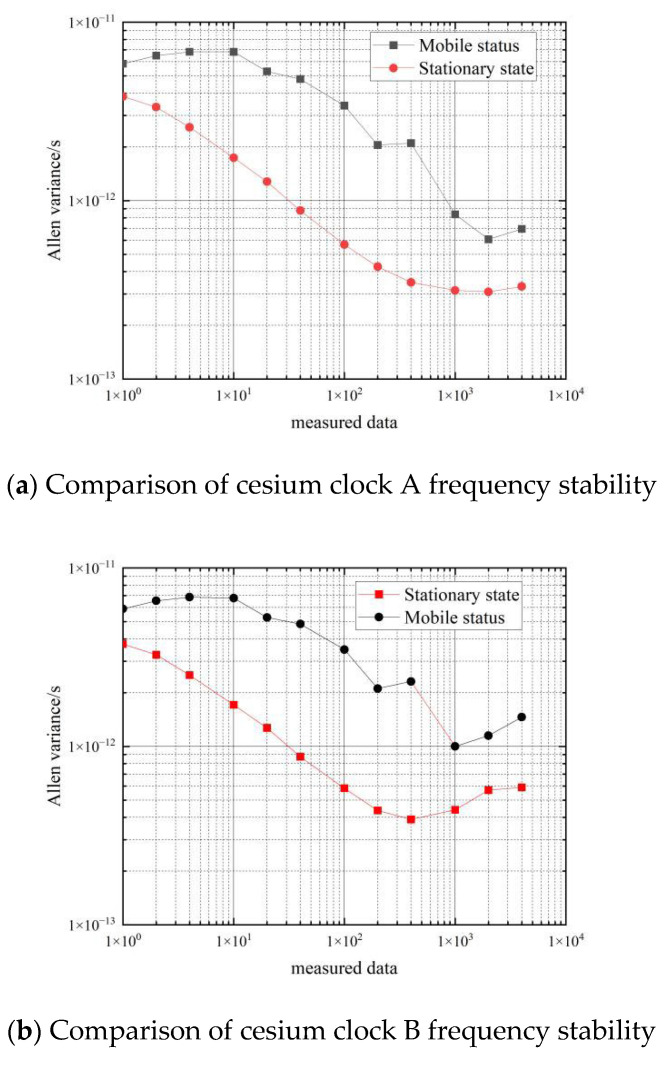
Comparison of the frequency stability of cesium clocks at a stationary and moving state.

**Figure 13 sensors-22-09886-f013:**
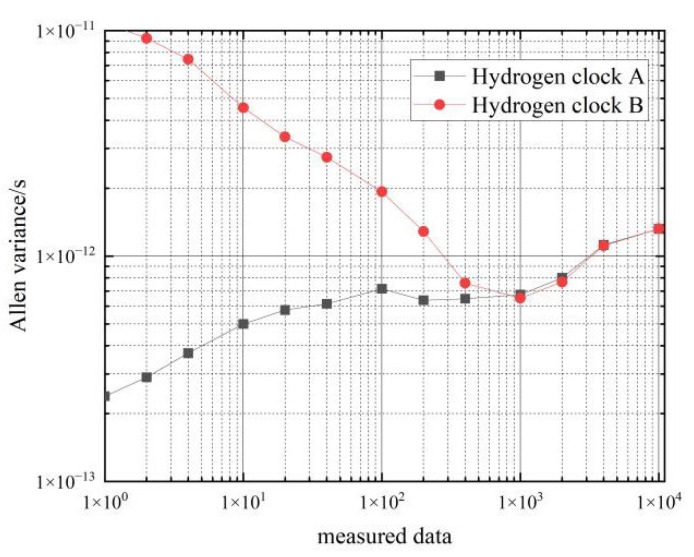
Frequency stability during the recovery phase of hydrogen clocks.

**Figure 14 sensors-22-09886-f014:**
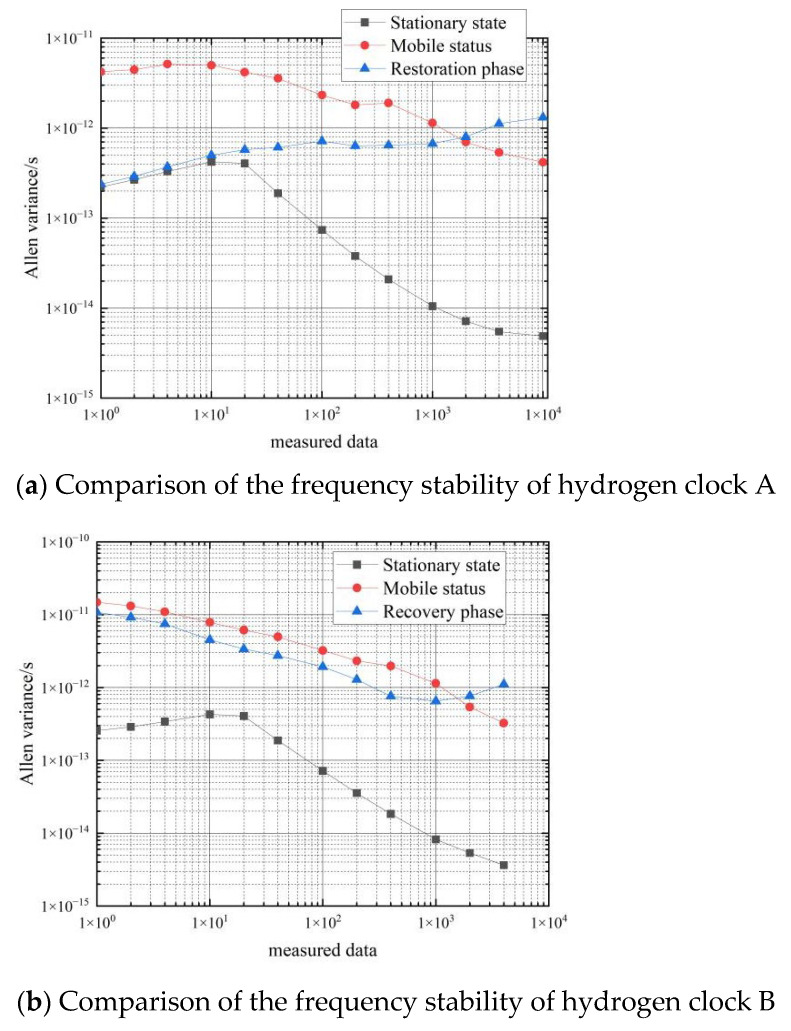
Comparison of the frequency stability of three states of the hydrogen clock.

**Figure 15 sensors-22-09886-f015:**
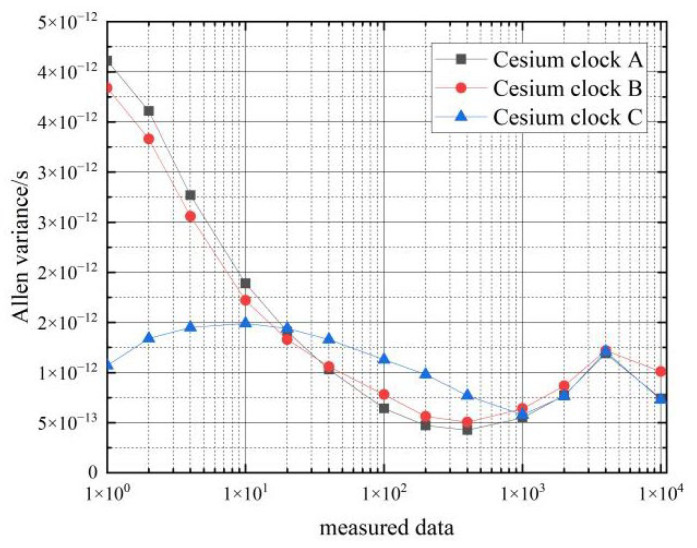
Frequency stability during the recovery phase of cesium clocks.

**Figure 16 sensors-22-09886-f016:**
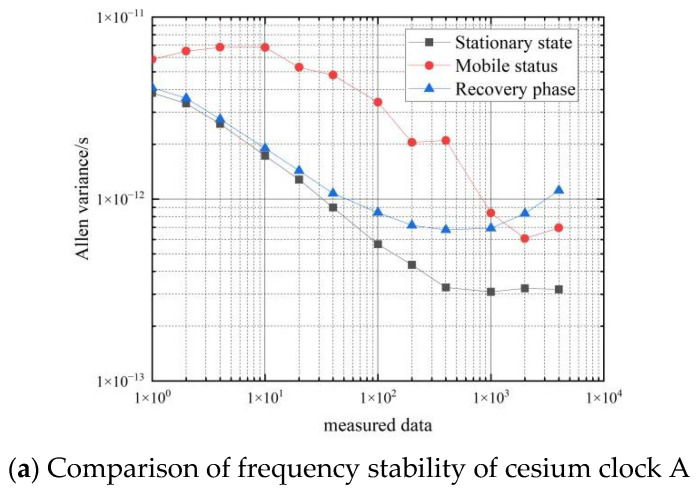
Comparison of frequency stability in the three states of the cesium clock.

**Figure 17 sensors-22-09886-f017:**
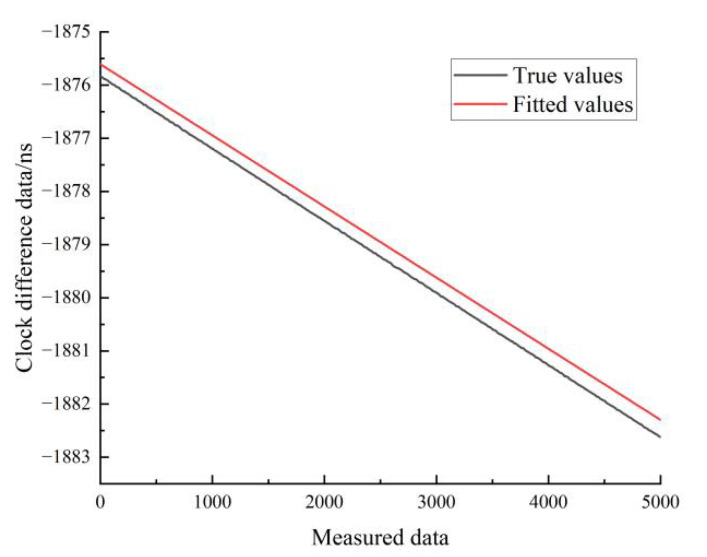
Comparison of linear fitted and true values for hydrogen clock A at stationary state.

**Figure 18 sensors-22-09886-f018:**
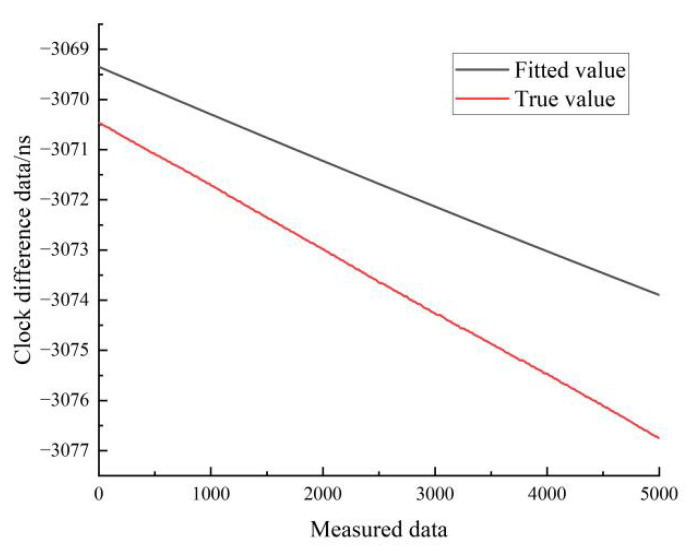
Comparison of linear fitted and true values for the moving state of hydrogen clock A.

**Figure 19 sensors-22-09886-f019:**
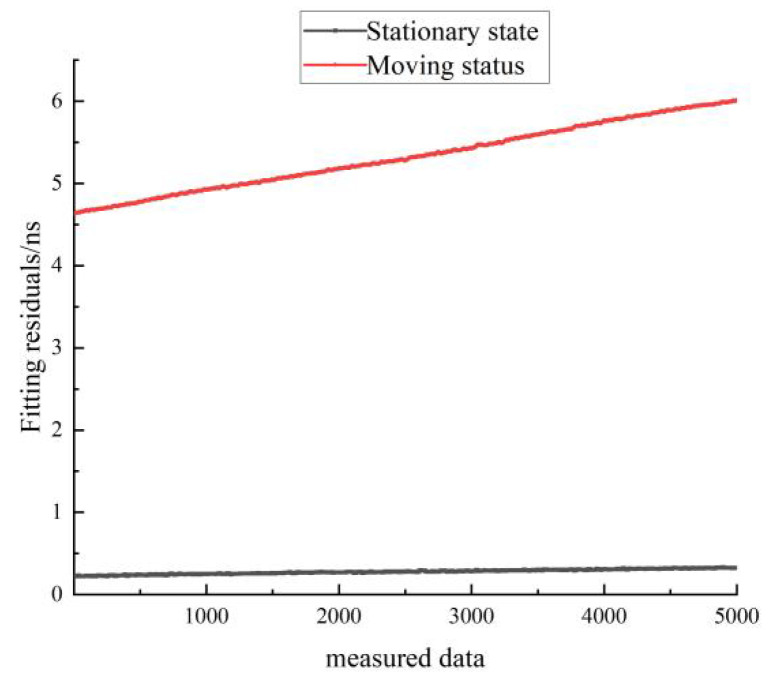
Comparison of the residuals of the linear fit for the stationary and moving states of hydrogen clock A.

**Figure 20 sensors-22-09886-f020:**
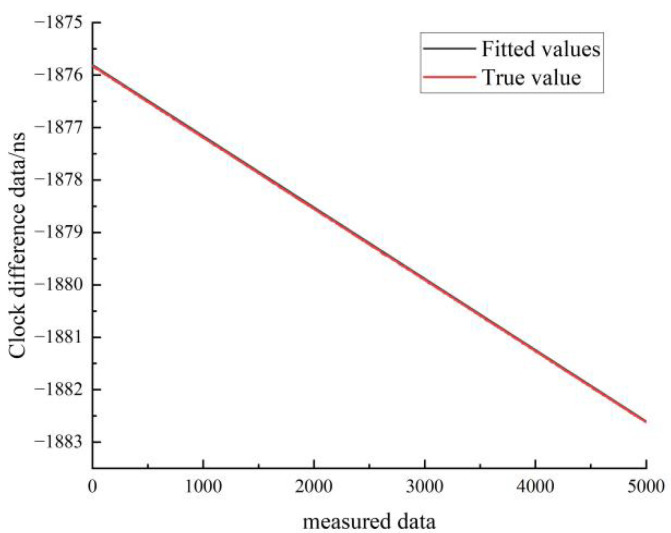
Comparison between the quadratic fit and the true value of the hydrogen clock A at stationary state.

**Figure 21 sensors-22-09886-f021:**
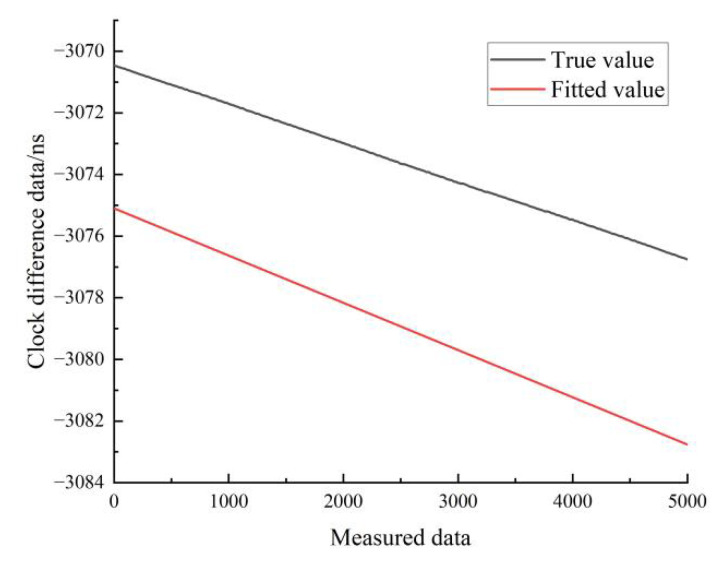
Comparison of the quadratic fitted and true values for the moving state of hydrogen clock A.

**Figure 22 sensors-22-09886-f022:**
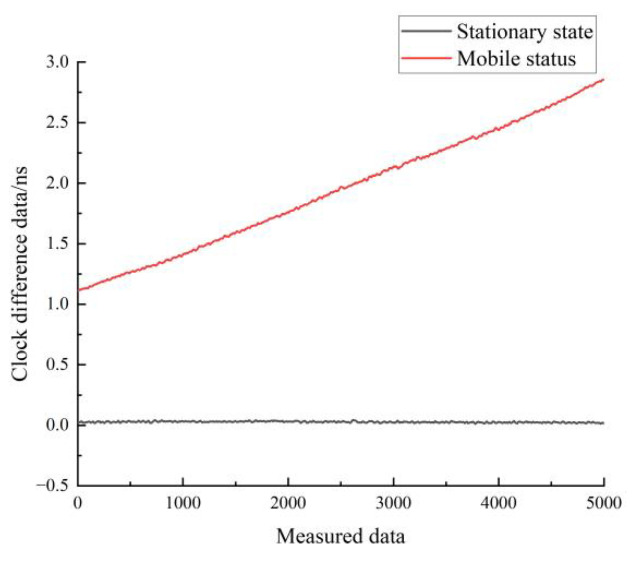
Comparison of the residuals of the quadratic fit for the stationary and moving states of the hydrogen clock A.

**Figure 23 sensors-22-09886-f023:**
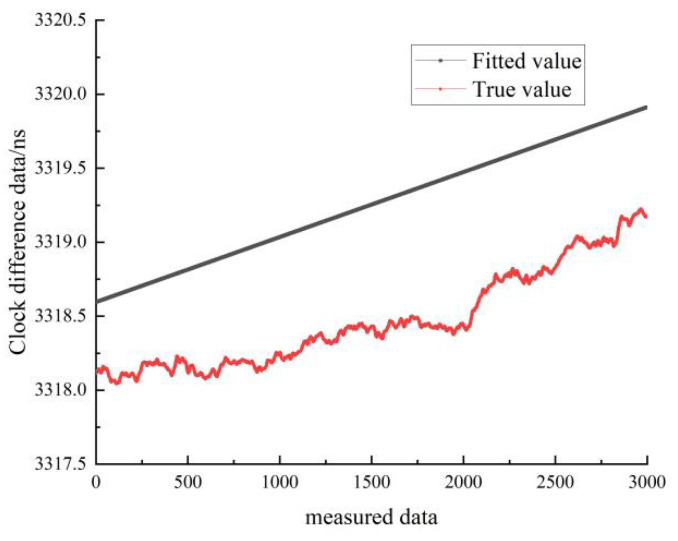
Comparison of the linearly fitted and true values of cesium clock A in the stationary state.

**Figure 24 sensors-22-09886-f024:**
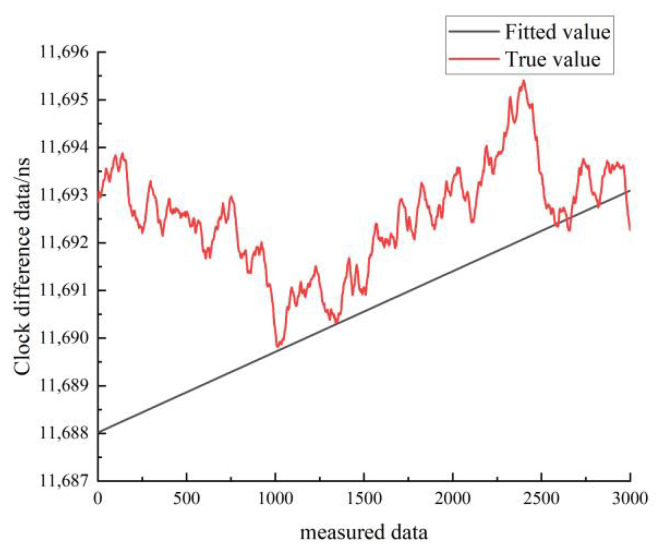
Comparison of linear fitted and true values for the moving state of the cesium clock A.

**Figure 25 sensors-22-09886-f025:**
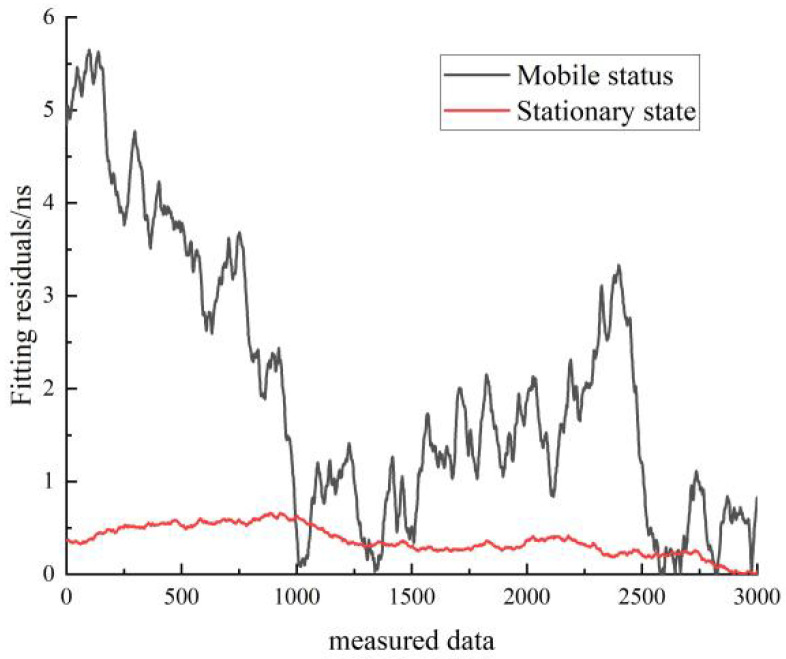
Comparison of the residuals of the linear fit for the stationary and moving states of the cesium clock A.

**Figure 26 sensors-22-09886-f026:**
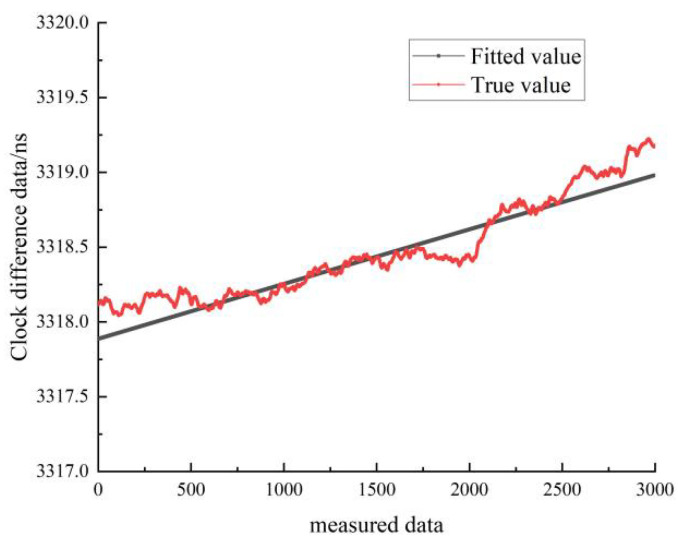
Comparison between the quadratic fit and the true value of cesium clock A at stationary state.

**Figure 27 sensors-22-09886-f027:**
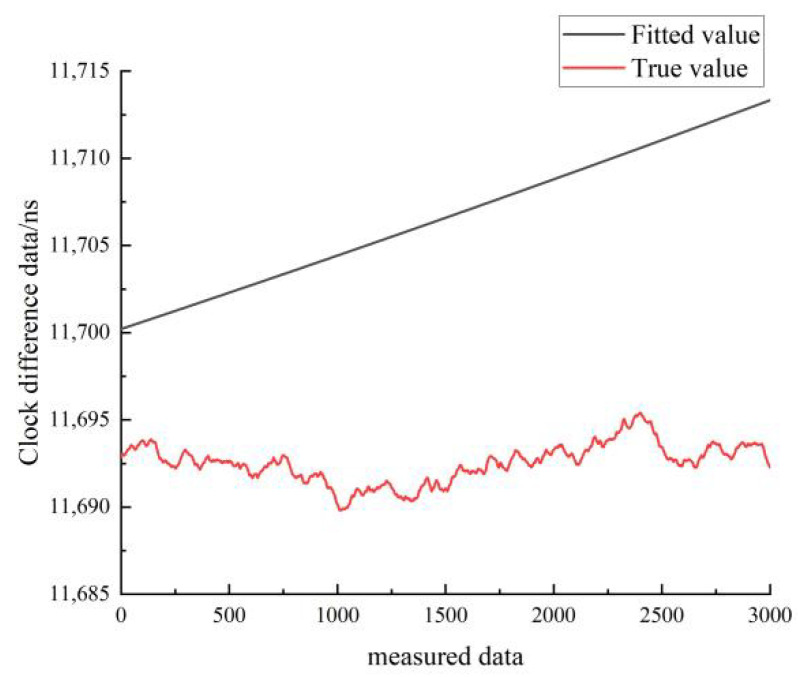
Comparison of the quadratic fitted and true values for the moving state of cesium clock A.

**Figure 28 sensors-22-09886-f028:**
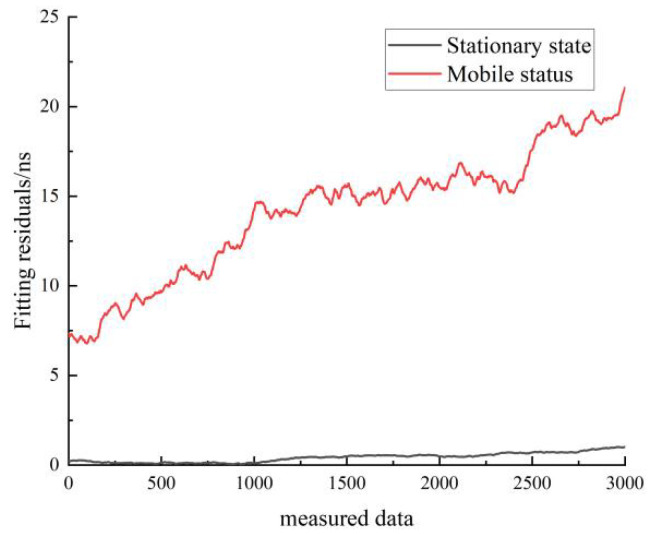
Comparison of the residuals of the quadratic fit for the stationary and moving states of cesium clock A.

**Table 1 sensors-22-09886-t001:** Nominal indexes of hydrogen atomic clock under static state.

Index Name	Index Requirements
Frequency stability of 1 s	≤$5\times 10^{-13}$
Frequency stability of 10 s	≤$1\times 10^{-13}$
Frequency stability of 100 s	≤$2\times 10^{-14}$
Frequency stability of 1000 s	≤$8\times 10^{-15}$
Frequency stability of 10,000 s	≤$5\times 10^{-15}$

**Table 2 sensors-22-09886-t002:** Calculations of frequency stability of hydrogen clocks at stationary state.

Measured Data/s	Car 1 Hydrogen Clock Mutual Evaluation/s
1	$1.87\times 10^{-13}$
10	$6.58\times 10^{-14}$
100	$1.28\times 10^{-14}$
1000	$4.29\times 10^{-15}$
10,000	$1.54\times 10^{-15}$

**Table 3 sensors-22-09886-t003:** Calculated frequency stability of each cesium clock at stationary state.

Measured Data	Car 1 Cesium Clock A/s	Car 1 Cesium Clock B/s	Car 1 Cesium Clock C/s	Car 2 Cesium Clock A/s	Car 2 Cesium Clock B/s	Car 2 Cesium Clock C/s
1	$3.83\times 10^{-12}$	$3.75\times 10^{-12}$	$1.08\times 10^{-12}$	$3.38\times 10^{-12}$	$3.69\times 10^{-12}$	$2.78\times 10^{-12}$
10	$1.74\times 10^{-12}$	$1.71\times 10^{-12}$	$1.50\times 10^{-12}$	$1.58\times 10^{-12}$	$1.70\times 10^{-12}$	$1.34\times 10^{-12}$
100	$5.67\times 10^{-13}$	$5.73\times 10^{-13}$	$9.77\times 10^{-13}$	$4.77\times 10^{-13}$	$5.71\times 10^{-13}$	$4.57\times 10^{-13}$
1000	$3.14\times 10^{-13}$	$4.65\times 10^{-13}$	$3.87\times 10^{-13}$	$5.77\times 10^{-13}$	$5.80\times 10^{-13}$	$7.23\times 10^{-13}$
10,000	$1.96\times 10^{-13}$	$4.46\times 10^{-13}$	$1.45\times 10^{-13}$	$1.14\times 10^{-12}$	$5.34\times 10^{-13}$	$5.41\times 10^{-13}$

**Table 4 sensors-22-09886-t004:** Calculated frequency stability of the two hydrogen clocks in the moving state for mutual evaluation.

Measured Data/s	Allan’s Variance for the Two Hydrogen Clocks Mutual Evaluation/s
1	$1.04\times 10^{-11}$
10	$3.69\times 10^{-12}$
100	$1.22\times 10^{-12}$
200	$8.49\times 10^{-13}$
400	$6.07\times 10^{-13}$
1000	$4.64\times 10^{-13}$
2000	$3.65\times 10^{-13}$
4000	$3.52\times 10^{-13}$

**Table 5 sensors-22-09886-t005:** Calculated frequency stability of each cesium clock in the moving state.

Measured Data/s	Allan’s Variance of Cesium Clock A/s	Allan’s Variance of Cesium Clock B/s	Allan’s Variance of Cesium Clock C/s
1	$5.85\times 10^{-12}$	$5.90\times 10^{-12}$	$8.00\times 10^{-12}$
10	$6.44\times 10^{-12}$	$6.45\times 10^{-12}$	$1.23\times 10^{-11}$
100	$2.90\times 10^{-12}$	$2.91\times 10^{-12}$	$5.32\times 10^{-12}$
1000	$1.21\times 10^{-12}$	$1.44\times 10^{-12}$	$1.46\times 10^{-12}$
4000	$5.91\times 10^{-13}$	$1.47\times 10^{-12}$	$3.14\times 10^{-13}$

**Table 6 sensors-22-09886-t006:** Calculated frequency stability of each hydrogen clock in the recovery phase.

Measured Data/s	Allan’s Variance of Hydrogen Clock A/s	Allan’s Variance of Hydrogen Clock B/s
1	$2.36\times 10^{-13}$	$1.08\times 10^{-11}$
10	$4.73\times 10^{-13}$	$5.55\times 10^{-12}$
100	$3.24\times 10^{-13}$	$1.93\times 10^{-12}$
1000	$6.48\times 10^{-13}$	$6.50\times 10^{-13}$
4000	$1.41\times 10^{-13}$	$1.32\times 10^{-13}$

**Table 7 sensors-22-09886-t007:** Calculated frequency stability of each cesium clock during the conversion phase.

Measured Data/s	Allan’s Variance of Cesium Clock A/s	Allan’s Variance of Cesium Clock B/s	Allan’s Variance of Cesium Clock C/s
1	$4.09\times 10^{-12}$	$3.85\times 10^{-12}$	$1.07\times 10^{-12}$
10	$1.90\times 10^{-12}$	$1.75\times 10^{-12}$	$1.50\times 10^{-12}$
100	$8.43\times 10^{-13}$	$9.60\times 10^{-13}$	$1.25\times 10^{-13}$
1000	$6.92\times 10^{-13}$	$7.75\times 10^{-13}$	$7.61\times 10^{-13}$
10,000	$1.24\times 10^{-12}$	$1.39\times 10^{-12}$	$1.22\times 10^{-12}$

**Table 8 sensors-22-09886-t008:** Comparison of the three main types of noise in the stationary and moving states of the hydrogen atomic clock.

	WFM/s	RWFM/s	WPM/s
Hydrogen clock A at stationary state	$7.5708\times 10^{-12}$	$4.0083\times 10^{-13}$	$5.5705\times 10^{-14}$
Hydrogen clock A in mobile state	$1.1415\times 10^{-9}$	$6.6964\times 10^{-12}$	$3.9208\times 10^{-10}$
Hydrogen clock B at stationary state	$1.7187\times 10^{-10}$	$4.8927\times 10^{-10}$	$2.3574\times 10^{-13}$
Hydrogen clock B in mobile state	$1.1400\times 10^{-9}$	$4.4475\times 10^{-11}$	$3.9156\times 10^{-10}$

**Table 9 sensors-22-09886-t009:** Comparison of the three main types of noise in the stationary and moving states of a cesium atomic clock.

	WFM/s	RWFM/s	WPM/s
cesium clock A at stationary state	$2.5686\times 10^{-11}$	$9.2796\times 10^{-16}$	$1.7645\times 10^{-14}$
cesium clock A in mobile state	$8.4164\times 10^{-10}$	$9.1099\times 10^{-16}$	$1.1563\times 10^{-10}$
cesium clock B at stationary state	$2.5338\times 10^{-11}$	$1.3244\times 10^{-15}$	$1.7406\times 10^{-13}$
cesium clock B in mobile state	$9.2440\times 10^{-10}$	$1.7285\times 10^{-15}$	$1.2700\times 10^{-10}$

**Table 10 sensors-22-09886-t010:** Residuals and RMS of linear and quadratic fits for stationary and moving states.

	Stationary State Linear Fit	Stationary State Quadratic Fit	Moving State Linear Fit	Moving State Quadratic Fit
RMS	0.2801	0.02819	5.3366	2.0098
Fitting residuals/ns	0.2787	0.0276	5.3218	1.9463

**Table 11 sensors-22-09886-t011:** Residuals and RMS for linear and quadratic fits to the cesium clock at stationary and moving states.

	Stationary State Linear Fit	Stationary State Quadratic Fit	Moving State Linear Fit	Moving State Quadratic Fit
RMS	0.7938	0.1107	4.0616	45.6283
Fitting residuals	0.7837	0.1346	3.5940	44.8563

## Data Availability

The data of this paper are stored in the database of the National Time Service Center, and we can provide them at any time if necessary.

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
