# Peer review of "Test and Analysis of Timekeeping Performance of Atomic Clock"

_sensors, 2022, doi:10.3390/s22249886_

Round 1

Reviewer 1 Report

Herein, the test and analysis of time keeping performance of Chinese atomic clock have been systematically conducted. The authors have completed interesting works and results. Overall, this work can be considered for publication after minor revision. The comments are listed below.

1.      The actual configuration of the set-up for hydrogen atomic clock should better be presented.

2.      The fonts in figures should be unified.

3.      The format of the tables should be modified.

4.      About the statics, the error bar should be paid attention. Related figure processing can be seen at similar reports such as Corrosion Science, (2022) 200 110231; Phys. Rev. Lett. 117, 143004 etc.

Author Response

Dear reviewer, thank you very much for your comments. Your comments will help me improve my paper. In response to your comments, I have made the following modifications.

  1. The nominal index of hydrogen atomic clock is given
  2. The font in the figure is changed to be consistent
  3. The chart format was modified as required
  4. It is a pity that the data given in this paper are Allen variance data, and error bars cannot be given when drawing

Reviewer 2 Report

performance of domestic hydrogen atomic clocks and cesium atomic clocks in the moving state. This work belongs to a new kind of work, which can provide valuable reference for the researchers of atomic clocks.

1. The last sentence of the article summary said that the predictability of atomic clocks in the moving state is very poor, and this conclusion needs to be cautious. The prediction of the atomic clock is mainly checked in the static state. The starting point of this paper can be experiment and analysis of the predictability in the moving state, but not evaluation.

2. The unit in the chart of predictability part is wrong, and the unit of clock error data should not be second.

3. Figure 7 does not fully show the main content of the article and does not show the work of predictability in the figure.

4. The graph of predictability needs to be further explained and improved, such as Figure 18 and Figure 19. It is recommended to draw the last 5000 data of training set and 5000 data of test set in the same graph, and draw the result of subtracting the predicted value and measured value.

Author Response

Dear reviewer, thank you very much for your comments. Your comments will help me improve my paper. In response to your comments, I have made the following modifications.

  1. We should be more careful in evaluating the predictability of the moving state of the atomic clock. The work in this part of this paper is more exploratory experiments
  2. Changed the unit of ordinate in some figures
  3. Improved the content of Figure 7
  4. Updated the predictable data and graph, and plotted the graph of residual data

Reviewer 3 Report

The manuscript is very useful as lab report for the researcher involved in this field.  The presentation of the graph is very poor. I recommend this for publication after  making the graph presentable.  Some of the graphs data is not visible, error bar are not shown, not nicely framed etc.

Manuscript can be accepted after minor corrections. My main concerns are as follows

1. The graphs are not well presented. e.g. Allen variance graph would be more useful if it is shown with grid as the y-axis is logarithmic.

2. Framing of the graph would be recommended

3. Fonts of the graph are not readable.

Author Response

Dear reviewer, thank you very much for your comments. Your comments will help me improve my paper. In response to your comments, I have made the following modifications.

  1. A grid plot was used to plot the Allen variance
  2. The chart is given according to the format requirements
  3. All block diagrams are drawn using visio, and data diagrams are drawn using origin, which is a readable version

Your comments are very helpful to the improvement of my article. Thank you again for your comments